# Associations between Four Diet Quality Indexes and High Blood Pressure among Adults: Results from the 2015 Health Survey of Sao Paulo

**DOI:** 10.3390/nu16050629

**Published:** 2024-02-24

**Authors:** Paula Victoria Felix, Jaqueline Lopes Pereira, Regina Mara Fisberg

**Affiliations:** Department of Nutrition, School of Public Health, University of São Paulo, Sao Paulo 01246-904, Brazil; paula.victoria@gmail.com (P.V.F.); jaque.lps@gmail.com (J.L.P.)

**Keywords:** diet quality, blood pressure, cross-sectional study, nutritional epidemiology

## Abstract

Several dietary quality indexes (DQIs) have been proposed to investigate adherence to a healthy diet. However, only a few studies have been conducted to investigate their association with high blood pressure (BP) in Brazil. In the present work, we examine the association between four established DQIs—2020 Healthy Eating Index (HEI-2020), Dietary Approaches to Stop Hypertension (DASH), Alternative Healthy Eating Index (AHEI), and Brazilian Healthy Eating Index (BHEI)—and high BP in a cross-sectional sample of Brazilian adults from the 2015 Health Survey of São Paulo with Focus on Nutrition. Based on two 24 h recalls adjusted for the within-person variation, higher HEI-2020 and BHEI total scores were inversely related to elevated BP (HEI-2020: OR 0.94, BHEI: OR 0.95). Individuals at the second quartile (OR 0.33) and the fourth quartile of BHEI (OR 0.35), as well as individuals with higher scores on dairy components (HEI-2020: OR 0.80, BHEI: OR 0.83, DASH: OR 0.75), and fruit components (AHEI: OR 0.82, HEI-2020: OR 0.72, BHEI: OR 0.77, DASH: OR 0.79) also presented lower odds for the occurrence of elevated BP. In conclusion, healthier diet quality using the HEI-2020 and BHEI indexes and the fruit and dairy components were identified as protective factors for high BP.

## 1. Introduction

The sustained rise in blood pressure (BP), also known as hypertension (HTN), is a leading cause of cardiovascular diseases (CVDs), disability, and premature death [1,2]. Even mild BP elevations manifesting as high-normal BP have been associated with cardiovascular risk [3]. A comprehensive analysis of hypertension prevalence and care in 184 countries showed that, from 1990 to 2019, the number of adults with HTN doubled worldwide, reaching 32% of the population, with most of the increase occurring in low- and middle-income regions [4]. Despite the known harmful effects of BP above recommended parameters, it is estimated that only one in five adults with HTN has it under control [5]. Yet, well-controlled BP has the potential to prevent clinical complications, enhance quality of life, and positively influence long-term prognosis [2,4].

BP levels are affected by several sociodemographic, environmental, and behavioral factors, including unhealthy diets (excessive salt intake, high intake of saturated and trans fats, low consumption of fruits and vegetables), tobacco use, harmful use of alcohol, physical inactivity, and obesity [5,6]. Interestingly, quality of diet is considered one of the main modifiable risk factors for HTN [6,7].

Due to the importance of assessing the nutritional quality of diets, several dietary quality indexes (DQIs) have been developed worldwide, which include the well-established 2020 Healthy Eating Index (HEI-2020) [8], the Alternate Healthy Eating Index (AHEI) [9], and the Dietary Approaches to Stop Hypertension (DASH) [10]. These tools represent a comprehensive means to determine a priori overall dietary patterns and condense the complexity of human diets into a single value, simultaneously considering population food guidelines and the interactions between nutrients, food preparation methods, and eating patterns [11].

Despite the wide variety of indexes used to measure diet quality, some concerns persist regarding their ability to predict cardiometabolic outcomes. For instance, inverse associations between HEI, AHEI, and DASH scores and some cardiometabolic risk factors have been documented in observational [12] and interventional studies [13,14]. However, these findings are mainly derived from high-income countries, and their association with BP presents controversial results [15,16,17,18,19].

The Longitudinal Study of Adult Health (ELSA-Brazil) showed that high adherence to the DASH diet was associated with a reduced risk of HTN over approximately 3.8 years of follow-up. This association was of borderline statistical significance after adjustment for body mass index (BMI), suggesting that body weight might play a relevant role in mediating the effects of the DASH diet on BP levels in the Brazilian population [17]. On the other hand, in the PREDIMED-Plus randomized trial, participants who showed the highest adherence, compared to the lowest, did not exhibit a reduced prevalence of HTN across any of the eight DQIs evaluated [20]. Similarly, no significant results were found between DQIs and BP after adjusting for confounding factors in Iranian [19] and Japanese adults [15].

Given that the structure of diets varies among countries, understanding the construction of DQI is also relevant, as differences in their underlying composition can result in variations in predictive power, making them potentially unsuitable for use in a specific population [11].

A frequently used index to evaluate diet quality in Brazil is the Brazilian Healthy Eating Index Revised (BHEI), adapted from the 2005 Healthy Eating Index [21]. While the relationship between BHEI and cardiometabolic outcomes is limited, it has been observed that higher BHEI scores are associated with lower chances of excess body weight and CDV risk factors among adolescents [22] and better lipid profiles among adults [23]. As far as we know, no studies were found associating it with BP, suggesting the need to assess the effectiveness of the BHEI in predicting this outcome, identify which component is more likely to influence this potential association, and compare it with other frequently used indexes. A deeper understanding of how well these DQIs relate to BP could enhance the efficacy of public health messages.

We hypothesize that higher DQIs scores are associated with lower chances of high BP and, further, that the BHEI exhibits the greatest magnitude among other indexes in this association. Therefore, this study aimed to examine the relationship between diet quality and BP in the population of Sao Paulo city.

## 2. Materials and Methods

### 2.1. Study Design and Population

The Health Survey of São Paulo (HSSP, Portuguese acronym “ISA-Capital”) is a household population-based, cross-sectional Brazilian study conducted in 2003, 2008, and 2015. It was designed to collect information about individuals living in urban areas of São Paulo city with respect to diet, lifestyle, acute and chronic morbidities, preventive practices, and use of health services by means of stratified sampling by clusters carried out in two stages (census tracts and households); more study details and sample design have been published elsewhere [24].

For the present study, we used a sub-sample of the 2015 HSSP survey, drawn to compose the Health Survey of São Paulo with Focus on Nutrition (“ISA-Nutrition”). Dietary data from 1742 individuals were collected. Adults and older adults (aged ≥ 19 years) with complete dietary data were selected (1235), of which 633 presented additional information on BP measurements (see Appendix A).

This study was prepared following the Strengthening the Reporting of Observational Studies in Epidemiology-Nutritional Epidemiology (STROBE-Nut) statement specified for nutritional epidemiologic investigations [25]. 

The present study was conducted following the principles outlined in the Declaration of Helsinki, the Brazilian Resolution Number 196/96 on research involving human subjects, and under Brazilian Law #5534 from 14 November 1968, which guarantees the confidentiality of the information collected by all national censuses and surveys. The 2015 ISA-Capital (protocol 36607614.5.0000.5421), as well as the present study (protocol 48960621.9.0000.5421), were approved by the Institutional Review Board of the School of Public Health, University of São Paulo. Written and verbal informed consent was obtained from all participants.

### 2.2. Sociodemographic and Anthropometric Information

Individual information on age, sex, self-reported ethnicity, educational level (years of schooling), smoking status (smoker or non-smoker), dietary habits, and per capita family income was gathered through a structured questionnaire conducted by trained interviewers in the household.

Self-reported ethnicity categorization was based on the Brazilian Institute of Geography and Statistics (IBGE) ethnic-racial classification as Black, Indigenous, Mixed, White, or Yellow [26]. For analysis purposes, it was categorized as White or Yellow and Black, Mixed, or Indigenous. Per capita family income was calculated by summing all monetary and non-monetary income reported by family members divided by the number of family members and categorized as less than one, from one to three, and three or more minimum wages, considering that the minimum wage was 954.00 Brazilian Real (BRL) in 2018 (equivalent to USD 298.53, 1 USD = 3.20 BRL on 15 January 2018). Smoking status was determined based on questions about current smoking, the number of cigarettes smoked daily, and whether one was a current smoker or a non-smoker.

Physical activity data were obtained through the validated “International Physical Activity Questionnaire” long version [27]. Leisure-time physical activity (e.g., walking, dancing, gardening, cycling, and swimming) was considered for this study. Participants were classified as meeting or not meeting the recommendations for physical activity (i.e., ≥150 vs. <150 min/week) according to the World Health Organization (WHO) guidelines [28].

Participants’ self-reported body weight and height were used to calculate BMI. Individuals were categorized into two groups: without overweight (BMI < 25 kg/m^2^ for adults [29] and BMI < 28 kg/m^2^ for older adults [30]) and overweight/obese (BMI ≥ 25 kg/m^2^ for adults and BMI ≥ 28 kg/m^2^ for older adults). The self-reported measure was previously validated in this population, presenting high sensitivity and specificity on that matter [31].

### 2.3. Blood Pressure Measurement

BP was measured according to the Seventh Brazilian Guidelines for Hypertension [32], using a validated automatic oscillometer (Omron^®^, model HEM-712 C, Omron Health Care, Inc., Vernon Hills, IL, USA) handled by a nursing technician, who also collected data on antihypertensive drug use. Measurements were conducted following a five-minute resting period in a seated position, with participants’ arms supported at heart level. BP was initially measured in the right arm, followed by a measurement in the left arm one minute later. An additional measurement was taken in the arm displaying the highest value. In cases where there was a discrepancy greater than 10% between the readings, a third measurement was obtained. The arithmetic mean of three assessments was recorded as the final BP. The participants were considered to have high BP if they had a systolic blood pressure ≥ 140 mmHg and/or a diastolic blood pressure ≥ 90 mmHg, according to national and international recommendations [32,33].

### 2.4. Dietary Data

The dietary intake was assessed using two non-consecutive 24 h dietary recalls (24 HR) collected on different days of the week, weekends, and seasons using the procedures of the Multiple Pass Method [34]. The Nutrition Data System for Research (NDS-R) software version 2021 (Nutrition Coordinating Center, University of Minnesota, Minneapolis, MN, USA) was used to estimate energy and nutrients collected from the 24 HR. As the NDS-R uses the United States Department of Agriculture food composition database, our study’s energy and nutrient values were compared to other Brazilian food composition databases [35,36]. Thus, any discrepancies were corrected for data processing.

The usual intake of nutrients and food groups consumed by the population was estimated by the National Cancer Institute method, which considers within- and between-person variance components and addresses the substantial intra-individual variation inherent in 24 HR [37].

### 2.5. Implausible Dietary Energy Intake

Individuals were categorized as plausible reporters, under-reporters, and over-reporters according to the implausible dietary energy intake classification. Usual energy intake was compared to the estimated energy requirements (EER) assessed by the Institute of Medicine equations stratified for sex and nutritional status [38,39]. The ratio between energy intake and energy requirements (EI/EER) was classified into the categories of energy misreporting according to the cut-off points determined by Huang et al. (2005) [40]. The plausible range of EI/EER in the population was 0.745–1.255 for adults and 0.734–1.266 for older adults. In this study, only individuals classified as plausible reporters and under-reporters were included in the regression model, as the number of over-reporters was very low (*n* = 3).

### 2.6. Dietary Quality Indexes

#### 2.6.1. 2020 Healthy Eating Index

The HEI-2020 was based on key recommendations from the 2020–2025 Dietary Guidelines for Americans [8], which comprise nine adequacy components (including total fruits, whole fruits, total vegetables, greens and beans, whole grains, dairy, total protein foods, seafood, and plant proteins, fatty acid ratio) and four moderation components (refined grains, sodium, saturated fats, and added sugars). For the adequacy components, subjects with the highest intake were assigned the highest score, while those with the lowest intake received the lowest score. In contrast, participants who consumed the highest amount of moderation components were assigned scores proportionally lower. The scoring algorithm operates based on density, with component scores summed to result in a total score ranging from 0 to 100.

#### 2.6.2. Alternative Healthy Eating Index 

The most recent version of the AHEI was developed by Chiuve et al. [9]. It is based on features of the original HEI and uses an absolute intake approach compared to a nutrient-density basis. The AHEI comprises 11 components: six favoring higher intake (vegetables, fruit, whole grains, nuts, and legumes, long chain omega-3 fatty acids, and polyunsaturated fatty acids), one component for which moderate intake is better (alcohol), and four components that must be limited or avoided (sugar-sweetened drinks and fruit juice, red and processed meat, trans fats, and sodium). Component scores are summed to a total AHEI score that ranges from 0 to 110 points.

#### 2.6.3. Dietary Approaches to Stop Hypertension

The DASH diet index developed by Fung et al. [10] is comprised of eight components based on foods and nutrients emphasized in the DASH eating guide, according to the National Heart, Lung, and Blood Institute [41]. The scoring system is based on quintile rankings; individuals receive a score from 1 (lowest quintile) to 5 (highest quintile) for intakes of vegetables, fruits and nuts, legumes, low-fat dairy products, and whole grains. In contrast, individuals receive a score from 1 (the highest quintile) to 5 (the lowest quintile) for intakes of sodium, red and processed meats, and sweetened beverages. The component scores were summed to yield the total DASH score, ranging from 8 to 40 points.

#### 2.6.4. Brazilian Healthy Eating Index Revised

The BHEI [21] was based on recommendations from the Food Guide for the Brazilian Population 2006 [42], the HEI-2005 [43], the WHO [44], and the guidelines of the Brazilian Society of Cardiology [45]. The BHEI is expressed in energy density, with a maximum score of 100, and comprises 12 components, including nine food groups (total fruit; whole fruit; total vegetables; dark green and orange vegetables; total grains; whole grains; milk and dairy; meat, eggs, and legumes; oils); two nutrients (saturated fat and sodium); and the last component quantifies the energy contribution from solid fats, alcohol, and added sugar (SoFAAS). Intermediate scores for each component were calculated proportionately.

Higher scores on the selected indexes indicate better adherence to the corresponding dietary recommendations and guidelines. The components of each DQI are listed in Appendix A.

### 2.7. Statistical Analyses

Descriptive analyses, including median, percentage, and interquartile range (IQR), were performed using Stata^®^ software (version 14.0, 2011, Stata Corp LP) taking into account the complex sampling design and significance level of 5%. The Theil–Sen median test for complex sampling design tested differences in socioeconomic, demographic, anthropometric, and lifestyle variables. The post hoc Dunn test was used to analyze the significance between groups. Data were presented considering a non-parametric distribution.

Spearman’s correlation method assessed the inter-correlation between the DQIs and their correlations with energy, macronutrient, and micronutrient intake. Stepwise logistic regression models, after controlling for confounding factors (e.g., age, sex, income status, BMI, antihypertensive drug use), were conducted to verify associations between diet quality and elevated BP. For the AHEI and DASH models, there was an additional adjustment for total energy intake. Results were presented as odds ratios (ORs) along with their respective 95% confidence intervals (95% CI). The calibration of each model was assessed using the Hosmer-Lemeshow goodness-of-fit test.

## 3. Results

### 3.1. Study Population Characteristics

The sociodemographic and lifestyle characteristics of the participants according to each DQI are shown in Table 1. The sample is predominantly composed of adults aged between 31 and 50 years old (35.2%), self-declared as White or Yellow (53.4%), with schooling up to high school (71.7%), and having per capita family income between 1 and 3 minimum wages (45.3%). Most of the population does not meet the recommendation of leisure time physical activity (81%), are non-smokers (83.5%), non-users of antihypertensive drugs (76.5%), and does not present excessive body weight (52.3%). The misreporting classification indicates a significant proportion of individuals categorized as under-reporters (58.6%).

The total population presented intermediate median values of diet quality for all DQIs evaluated. Among the characteristics investigated, young adults (19–30 years old) had the lowest diet quality scores, and elderly individuals >70 years old had the higher diet quality scores.

Women, individuals classified as White or Yellow, who reported antihypertensive drug use, and under-reporters presented higher diet quality on all indexes. Individuals with higher per capita income scored better on the AHEI, HEI-2020, and DASH. Individuals with schooling up to high school had higher scores on the BHEI. According to HEI-2020, BHEI, and DASH, better diet quality was found in non-smokers. Physical activity level and BMI status did not significantly differ in diet quality.

### 3.2. Nutrient Correlations

The AHEI, HEI-2020, BHEI, and DASH dietary scores are strongly correlated, with Spearman’s correlation coefficients ranging from 0.65 to 0.89 in the pooled data (Table 2). Higher index scores are directly correlated with protein intake, fiber, calcium, potassium, vitamins A, C, D, and E, and inversely correlated with total energy, total grams of foods and beverages, added sugar, and sodium. The DQIs differed concerning carbohydrates and total fat consumption; total fat showed a negative correlation with HEI-2020 and BHEI and a positive correlation with AHEI. Carbohydrates showed a negative correlation only with HEI-2020.

### 3.3. Association between DQIs and BP

After controlling for potential covariates (age, sex, BMI, misreporting classification, physical activity, use of antihypertensive drugs, per capita family income, self-declared skin color-race, daily energy intake), higher HEI-2020 and BHEI total scores were inversely associated with elevated BP (HEI-2020: OR 0.94, 95% CI 0.89, 0.99; BHEI: OR 0.95, 95% CI 0.91, 0.99). Individuals at the second quartile (OR 0.33, 95% CI 0.12, 0.90) and the fourth quartile of BHEI (OR 0.35, 95% CI 0.13, 0.89) also presented lower odds for the occurrence of elevated BP (Table 3). No significant association was found between DASH and AHEI and the odds of high BP.

Regarding the individual components of each DQI, participants with higher scores on dairy components (HEI-2020: OR 0.80, 95% CI 0.69–0.92; BHEI: OR 0.83, 95% CI 0.72–0.96; DASH: OR 0.75, 95% CI 0.59–0.97) and fruit components (AHEI: OR 0.82, 95% CI 0.67–0.99; HEI-2020: OR 0.72, 95% CI 0.55–0.91; BHEI: OR 0.79, 95% CI 0.61–0.98; DASH: OR 0.80, 95% CI 0.63–0.99) had a lower occurrence of elevated BP (Table 4).

## 4. Discussion

Among the four selected a priori-defined DQIs, higher adherence to the HEI-2020 and the BHEI was associated with lower odds of having high BP, while no significant results were observed for the AHEI and DASH indexes. Analysis of individual components across all indexes revealed that higher intakes of fruits and dairy were associated with a protective effect against the development of high BP. These results were irrespective of several sociodemographic and lifestyle factors, including age, sex, BMI, misreporting classification of energy intake, per capita family income, self-declared skin color-race, physical activity, total energy intake, and the use of antihypertensive drugs.

Individuals in the second and fourth quartiles of BHEI had significantly lower odds of having high BP compared to those in the first quartile. This association was found only in the BHEI score, indicating that using a population-specific diet quality index may provide a clearer representation of the association between diet quality and BP outcomes. This finding aligns with a study including adolescents from Sao Paulo that identified an association between higher diet quality and lower chances of CDV risk factors using the BHEI but not with the AHEI scores [22].

Often, indexes developed for specific populations, such as the HEI or AHEI in the United States, are applied to different populations without undergoing thorough validation. This practice is driven by the inclusion of key nutrients and foods with recognized health effects, creating the impression that the tool is culturally neutral [46]. However, subtle variations in dietary practices can attenuate the association between diet quality, as assessed by non-population-specific indexes, and health outcomes. Ideally, existing indexes could serve as models for creating a local DQI tailored to the actual circumstances of each country. This adaptation and subsequent validation for each population would enhance the accuracy of the assessment [11].

In Brazil, according to the 2021 Chronic Disease Risk and Protective Factors Surveillance Telephone Survey (VIGITEL), the prevalence of self-reported HTN among the adult population was 26.3% [47], similar to that found in countries such as Australia (29.3%), Canada (22.1%), and China (27.3%) [48]. Several studies have explored the association between DQIs and BP, yielding varied findings. Some studies have reported a protective effect of the HEI diet on BP or the risk of HTN [12,16,49]; however, not all investigations have found this connection [19].

Regarding BHEI, no previous studies evaluated this index or HNT or BP. However, data from the ISA-Nutrition 2015 showed that adolescents could lower the odds of excess body weight by 13% and CDV intermediate factors by 11% with each additional unit increase in BHEI [22]. Furthermore, Fujii et al. (2019) demonstrated that the BHEI evaluation and the genetic risk score could be valuable tools to predict cardiometabolic risk in São Paulo adults [23]. In the present study, Brazilian adults could lower the odds of high BP by 5% with each additional unit increase in BHEI and HEI-2020 scores. This is particularly important because small reductions in BP may have significant public health effects.

There was no association between AHEI, DASH, and BP. This finding contrasts with previous results from systematic reviews and meta-analyses of randomized controlled trials [50] and observational studies [51], which demonstrated that adherence to the DASH diet was accompanied by significant BP reduction. Similar beneficial associations with the DASH score were also observed in prospective studies [12,17,52]. On the other hand, some studies found no association [18,20] or reported that this association disappeared after further adjustment for confounders [19].

The diets of these individuals align more closely with the BHEI than with indexes developed in other countries. However, given the wealth of evidence for the health benefits of a DASH diet, the absence of association observed in the current study is noteworthy. This evidence illustrates the diversity in the association between DQIs and metabolic outcomes across populations, which could be related to the differences in the sample size, different designs of the studies, and methods used to assess dietary intake. It is also possible that the creation of the composite scores may not have captured the relative impact among the food groups or nutrients entirely as they relate to BP in the São Paulo population [53].

Further analysis of the individual components of each DQI was conducted to determine possible dietary components that might affect BP. The intake of dairy and fruit components within recommendations was associated with a 17–28% reduction in high BP in adults. The consumption of fruits and dairy products has been intensively investigated in the literature, although the exact mechanisms by which these compounds impact BP are not entirely elucidated. It seems that fruit consumption is beneficial due to the presence of flavonoids, carotenoids, and high contents of potassium, magnesium, vitamin C, and folic acid, which have been postulated to improve endothelial function, modulate baroreflex sensitivity, and increase antioxidant activity, thereby potentially lowering BP [54,55,56].

Likewise, there is evidence that dairy food intake may affect BP, considering that they are rich sources of micronutrients, vitamin D, and bioactive peptides. These components participate in the regulation of vascular resistance, promoting vasodilation through increased nitric oxide production, reducing renal sodium retention, improving insulin sensitivity, and preventing blood vessel constriction [57,58].

The lack of association between vegetable consumption and BP in this study may be related to the types of vegetables consumed and their cooking methods. Processing vegetables can alter their nutritional and chemical composition, while the variety of vegetables consumed may affect BP differently, possibly leading to a deviation of effect [59]. Notwithstanding, promoting vegetable consumption should persist, as its benefits extend beyond regulating BP.

The potential health implications of high sugar-sweetened beverage consumption encompass a range of issues, including excess body weight, type 2 diabetes, and dental problems [60]. A prior study conducted with the same population revealed that larger portions of soft drinks were correlated with increased body weight, with around 30% of the population consuming them [61]. Nevertheless, specific components directly related to sugar-sweetened beverages, like “Sugar-sweetened beverages and fruit juice” in the AHEI and “Sugar-sweetened beverages” in the DASH, as well as indirectly related factors like “Added Sugars” in the HEI-2020 and “SoFAAS” in the BHEI, did not exhibit an association with high BP in the current study.

Although it is recognized that excessive sodium intake leads to an increased risk of cardiovascular diseases [62], the present study does not rely on a direct biomarker such as 24 h sodium urine excretion. Instead, self-reported dietary methods are used, which often underestimate sodium intake due to the difficulty in quantifying added salt in food preparations and the sodium content of food items included in food databases [63]. This could explain the non-significant results when investigating the sodium component score and the isolated components of each DQI.

Some socio-demographic and lifestyle variables presented differences in the overall score for diet quality, including age group, sex, self-reported ethnicity, per capita family income, smoking status, and misreporting categories. These findings align with evidence showing that diet quality and eating behaviors are influenced by several factors embedded in socioeconomic and cultural contexts, lifestyle, and health behaviors [64].

Diet quality remained well short of the minimum guidelines, particularly among young adults, who showed the lowest scores for diet quality. The literature has consistently reported that a substantial proportion of young adults fail to adhere to national guidelines for healthy eating and tend to be less concerned with the negative impacts of unhealthy food [64,65]. Still, their food choices may forecast future health issues as they age and should be the target of initiatives aimed at providing adequate diet quality.

Conversely, older adults had the highest scores, but improvements are still needed to prevent complications resulting from high BP. From 2008 to 2015, São Paulo experienced a rise in the prevalence of intermediate factors of CVDs, where HTN exhibited the second-largest increase among the factors evaluated, with a 1.3-fold rise, surpassed only by diabetes [66]. This surge in healthcare demands presents a substantial challenge, calling for strategic planning in both health and economic public policies.

Correlations across all DQIs and essential macro- and micronutrients ranged from 0.06 to 0.811 in the present study and were significantly associated with most nutrients in the expected direction. Nutrients considered protective for developing CDV, such as potassium, fibers, and vitamins, were positively associated, while risk nutrients showed negative associations. This implied that these indexes share some similarities, e.g., promoting low intakes of saturated fat, sodium, and sugar-sweetened beverages with high intakes of fruit and vegetables, thus indicating an underlying similar dietary pattern. Our results also demonstrate that food groups sharing similar compositions can manifest variations in their weighting depending on the DQI, underscoring the importance of analyzing diets from a broad perspective.

These results corroborate scientific evidence endorsing actions to reduce population BP—a highly impactful measure to promote global public health [2]. Improving diet quality not only has no harmful side effects but also contributes to overall cardiovascular health and can reduce the requirement for BP-lowering medications [32]. On this basis, it is necessary to invest in a multidimensional approach to guide the population regarding diet quality, emphasizing the importance of greater consumption of healthy foods, including fruits and dairy products, as part of the daily preventive practices to be taken to prevent HTN.

As far as we know, this is the first study to evaluate the association between four dietary indexes and BP measurements in a sample of free-living adults and older adults from São Paulo, the largest city in Brazil. It is worth noting that statistical techniques were used to adjust for potential confounding factors, and this will aid in interpreting subsequent projects that examine associations between these dietary indexes and health outcomes in the ISA-Nutrition.

Despite the strength of this study, some methodological features should be considered. First, BP measurements were performed during a single visit, which may not fully capture the dynamic nature of individual BP fluctuations. Second, the dietary intake data relied on self-reported 24 HR, introducing potential random and systematic errors such that subjects might under- or overestimate their food consumption, resulting in misclassification of energy intake [67]. To address these concerns, 24 HR were collected by trained interviewers using standardized methods to reduce recall bias. Moreover, statistical modeling incorporated into the National Cancer Institute method was used to account for intra-individual variation in food consumption [24,34], and the models were further adjusted for the underreporting of energy intake, thereby improving the accuracy of the analyses.

## 5. Conclusions

Healthier diet quality is associated with lower odds of high BP in adults and older adults from São Paulo, Brazil. The application of a population-specific DQI has enhanced our ability to portray the nuanced relationship between diet quality and BP, providing further evidence backing the importance of considering regional differences in selecting a DQI. Through an evaluation of the isolated components of these indexes, the results emphasize the significance of promoting the consumption of fruits and dairy as a straightforward public health message aimed at reducing the burden of high BP. 

## Figures and Tables

**Table 1 nutrients-16-00629-t001:** Socioeconomic and lifestyle characteristics of the adult and older adult population in ISA-Nutrition (2015) according to diet quality indexes ^1^.

	ISA-Nutrition 2015														
Population Characteristics	Total Population			AHEI (0–110)			HEI-2020 (0–100)			BHEI (0–100)			DASH (8–40)		
	*n*	%	95% CI	Median	IQR	*p* ^3^	Median	IQR	*p*	Median	IQR	*p*	Median	IQR	*p*
Total population	1235	-	-	50.1	(46.1, 54.6)		56.8	(53.0, 61.4)		70.3	(65.3, 74.4)		25.0	(21.0, 29.0)	
Age group, years															
19–30	286	25.4	(22.3, 38.6)	45.5	(41.8, 48.9)		51.4	(49.1, 54.0)		65.0	(61.1, 68.9)		19.0	(17.0, 22.0)	
31–50	314	35.2	(31.7, 38.6)	48.8	(45.5, 52.0)		55.8	(53.2, 58.4)		69.1	(65.6, 72.9)		24.0	(21.0, 26.0)	
51–70	434	28.9	(25.9, 32.0)	53.4	(49.7, 57.2)		60.6	(58.1, 63.6)		73.6	(70.0, 75.9)		28.0	(25.0, 30.0)	
>70	201	10.6	(08.9, 12.4)	59.2	(56.2, 61.4)	<0.001	64.8	(61.8, 67.6)	<0.001	76.2	(74.1, 78.4)	<0.001	31.0	(30.0, 33.0)	<0.001
Sex															
Male	579	49.5	(46.0, 53.1)	46.6	(43.3, 50.4)		55.7	(52.0, 59.8)		69.6	(65.0, 74.2)		23.0	(19.0, 26.0)	
Female	656	50.5	(46.9, 54.0)	53.1	(49.8, 57.7)	<0.001	58.3	(53.7, 62.9)	<0.001	70.7	(66.0, 74.8)	0.032	27.0	(23.0, 30.0)	<0.001
Self-reported ethnicity															
White or Yellow	654	53.4	(49.4, 57.3)	51.1	(47.2, 56.1)		58.1	(53.6, 62.3)		71.0	(66.0, 74.9)		26.0	(21.0, 30.0)	
Black, Mixed, or Indigenous	571	46.7	(42.7, 50.6)	49.0	(45.1, 53.3)	<0.001	55.8	(51.9, 60.2)	<0.001	69.7	(64.6, 74.1)	0.024	24.0	(20.0, 27.0)	<0.001
Education level															
≤11 years of schooling (up to high school)	961	71.7	(67.5, 75.5)	50.1	(46.1, 54.7)		57.1	(53.0, 61.2)		71.0	(66.0, 74.8)		25.0	(21.0, 28.0)	
>11 years of schooling (above high school)	270	28.3	(24.5, 32.5)	50.2	(46.1, 54.5)	0.965	56.6	(52.8, 61.7)	0.909	68.6	(64.0, 72.7)	<0.001	25.0	(21.0, 29.0)	0.671
Per capita family income ^2^															
≤1 minimum wage	449	40.0	(35.4, 44.8)	49.1	(45.2, 53.1)		55.6	(51.8, 59.6)		69.4	(65.0, 74.1)		23.0	(20.0, 27.0)	
1–3 minimum wage	486	45.3	(40.9, 49.8)	49.9	(45.9, 54.4)		57.0	(53.2, 61.4)		70.4	(65.4, 74.3)		25.0	(21.0, 28.0)	
>3 minimum wage	145	14.6	(11.3, 18.7)	52.9	(48.6, 57.7)	<0.001	59.6	(54.8, 64.3)	<0.001	71.5	(66.2, 74.9)	0.065	28.0	(24.0, 31.0)	<0.001
Leisure time physical activity level															
Do not meet the recommendation	1015	81.0	(78.1, 83.6)	50.4	(46.4, 54.8)		57.0	(53.1, 61.4)		70.4	(65.5, 74.5)		25.0	(21.0, 29.0)	
Meet the recommendation	220	19.0	(16.4, 21.9)	49.7	(44.7, 53.6)	0.057	56.4	(52.0, 61.0)	0.186	69.7	(64.7, 74.2)	0.221	24.0	(20.0, 28.0)	0.142
Actual smoking status															
Non-smoker	1037	83.5	(80.9, 85.7)	50.3	(46.1, 55.0)		57.2	(52.9, 61.7)		70.6	(65.7, 74.6)		25.0	(21.0, 29.0)	
Current smoker	194	16.5	(14.3, 19.1)	49.0	(45.8, 53.9)	0.086	55.9	(53.3, 59.9)	0.013	68.5	(63.5, 75.5)	<0.001	24.0	(21.0, 27.0)	<0.001
Body weight status															
Without excess body weight	662	52.3	(48.9, 55.7)	49.8	(45.5, 54.7)		56.6	(52.1, 61.7)		70.1	(65.1, 74.5)		24.0	(20.0, 29.0)	
With excess body weight	548	47.7	(44.3, 51.1)	50.3	(46.5, 54.4)	0.502	57.1	(53.5, 61.1)	0.220	70.4	(65.9, 74.3)	0.676	25.0	(21.0, 28.0)	0.580
Antihypertensive drug use															
No	871	76.5	(73.6, 79.1)	43.9	(40.2, 47.7)		55.5	(51.9, 59.8)		68.9	(64.3, 73.2)		23.0	(20.0, 27.0)	
Yes	361	23.5	(20.9, 26.4)	50.4	(45.7, 54.2)	<0.001	61.6	(58.3, 64.9)	<0.001	74.2	(71.1, 76.8)	<0.001	29.0	(26.0, 32.0)	<0.001
Misreporting															
Plausible reporter	499	41.4	(38.9, 44.7)	48.2	(43.8, 53.1)		55.9	(51.9, 60.8)		69.5	(64.4, 74.2)		23.0	(20.0, 28.0)	
Under-reporter	711	58.6	(55.3, 61.9)	51.3	(47.5, 55.8)	<0.001	57.6	(53.5, 61.7)	0.001	70.6	(66.0, 74.6)	0.030	25.0	(22.0, 29.0)	<0.001

Abbreviations: AHEI, Alternative Healthy Eating Index; HEI-2020, Healthy Eating Index 2020; BHEI, Brazilian Healthy Eating Index Revised; DASH, Dietary Approaches to Stop Hypertension. ^1^ All the analyses considered the sampling survey design. ^2^ One MW was approximately USD 236 in 2015. ^3^ Median and interquartile ranges (IQRs) are described, and differences were evaluated using the Theil–Sen test. Post hoc Dunn’s test was applied to compare variables with three or more groups. All medians in the same variable are significantly different (*p* < 0.01).

**Table 2 nutrients-16-00629-t002:** Spearman’s correlation coefficients (r) among the diet quality indexes and essential macronutrients and micronutrients, ISA-Nutrition 2015.

	Total Population (*n* = 1235)			
	AHEI	HEI-2020	BHEI	DASH
HEI-2020	0.801 ***			
BHEI	0.654 ***	0.796 ***		
DASH	0.887 ***	0.895 ***	0.717 ***	
Total energy (kcal/d)	−0.743 ***	−0.481 ***	−0.327 ***	−0.610 ***
Total grams of foods and beverages (g/day)	−0.631 ***	−0.359 ***	−0.269 ***	−0.483 ***
Protein (%kcal)	0.227 ***	0.284 ***	0.293 ***	0.257 ***
Carbohydrates (%kcal)	0.028	−0.116 **	0.009	−0.014
Total Fat (%kcal)	0.068 *	−0.119 ***	−0.188 ***	−0.042
Total fiber (g/1000 kcal)	0.525 ***	0.603 ***	0.604 ***	0.577 ***
Added sugar (%kcal)	−0.415 ***	−0.599 ***	−0.642 ***	−0.488 ***
Sodium (mg/1000 kcal)	−0.094 ***	−0.090 ***	−0.104 ***	−0.089 ***
Calcium (mg/1000 kcal)	0.522 ***	0.593 ***	0.371 ***	0.610 ***
Potassium (mg/1000 kcal)	0.713 ***	0.806 ***	0.657 ***	0.773 ***
Vitamin A (mg/1000 kcal)	0.769 ***	0.689 ***	0.455 ***	0.763 ***
Vitamin C (mg/1000 kcal)	0.743 ***	0.805 ***	0.591 ***	0.811 ***
Vitamin D (mg/1000 kcal)	0.651 ***	0.587 ***	0.430 ***	0.619 ***
Vitamin E (mg/1000 kcal)	0.606 ***	0.502 ***	0.348 ***	0.563 ***

Abbreviations: AHEI, Alternative Healthy Eating Index; HEI-2020, Healthy Eating Index 2020; BHEI, Brazilian Healthy Eating Index Revised; DASH, Dietary Approaches to Stop Hypertension. Values were significantly different: * *p* < 0.05, ** *p* < 0.01, *** *p* < 0.001.

**Table 3 nutrients-16-00629-t003:** Association of diet quality indexes according to high blood pressure in ISA-Nutrition (2015) assessed using logistic regression models.

Quartile (Range of Scores)	Adjusted Model for High Blood Pressure (*n* = 633) ^1^		
	OR	95% CI	*p* *
AHEI (continuous)	0.941	(0.88, 1.00)	0.079
Q1 (ref)			
Q2	0.566	(0.19, 1.60)	0.284
Q3	0.365	(0.11, 1.12)	0.078
Q4	0.339	(0.09, 1.20)	0.094
HEI-2020 (continuous)	0.943	(0.89, 0.99)	0.043
Q1 (ref)			
Q2	0.580	(0.16, 2.09)	0.408
Q3	0.560	(0.16, 1.99)	0.372
Q4	0.276	(0.07, 1.05)	0.070
BHEI (continuous)	0.949	(0.91, 0.99)	0.059
Q1 (ref)			
Q2	0.333	(0.12, 0.90)	0.031
Q3	0.417	(0.16, 1.05)	0.064
Q4	0.346	(0.13, 0.89)	0.028
DASH (continuous)	0.942	(0.87, 1.01)	0.123
Q1 (ref)			
Q2	1.107	(0.29, 3.89)	0.879
Q3	0.877	(0.22, 3.45)	0.851
Q4	0.556	(0.12, 2.41)	0.434

Abbreviations: AHEI, Alternative Healthy Eating Index; HEI-2020, Healthy Eating Index 2020; BHEI, Brazilian Healthy Eating Index Revised; DASH, Dietary Approaches to Stop Hypertension. ^1^ All the models were adjusted for age (years), age squared (years^2^), sex, BMI (kg/m^2^), misreporting (plausible or underreported), per capita family income, self-declared skin color-race (White, Yellow or Black, Mixed, Indigenous), leisure-time physical activity (meet or do not meet WHO recommendation), and use of antihypertensive drugs. The AHEI and DASH models were also adjusted for total energy intake. * A *p*-value < 0.05 was considered statistically significant.

**Table 4 nutrients-16-00629-t004:** Association of diet quality components according to high blood pressure in ISA-Nutrition (2015) assessed using logistic regression models.

DQI Components	Adjusted Model for High Blood Pressure (*n* = 633) ^1^		
	OR	95% CI	*p* *
AHEI			
Whole fruits	0.823	(0.67, 0.99)	0.047
Total vegetables	1.022	(0.73, 1.41)	0.893
Whole grains	0.813	(0.52, 1.26)	0.358
Red and processed meat	1.033	(0.70, 1.52)	0.869
Nuts	-		
Long-chain (*n*-3) fats	1.061	(0.83, 1.34)	0.625
Polyunsaturated fatty acids	0.925	(0.73, 1.16)	0.504
Trans fat	0.996	(0.60, 1.64)	0.991
Sugar-sweetened beverages and fruit juice	0.911	(0.80, 1.02)	0.124
Sodium	0.975	(0.78, 1.21)	0.823
Alcohol	0.912	(0.73, 1.16)	0.529
HEI-2020			
Total Fruits	0.716	(0.55, 0.91)	0.009
Whole Fruits	0.819	(0.62, 1.07)	0.152
Total Vegetables	1.204	(0.83, 1.73)	0.318
Greens and Beans	0.635	(0.13, 2.89)	0.559
Whole Grains	0.768	(0.52, 1.11)	0.168
Dairy	0.802	(0.69, 0.92)	0.002
Total Protein Foods	0.804	(0.27, 2.35)	0.692
Seafood and Plant Proteins	1.397	(0.89, 2.17)	0.138
Fatty Acids	1.115	(0.96, 1.28)	0.129
Refined Grains	0.969	(0.85, 1.09)	0.615
Sodium	0.991	(0.85, 1.14)	0.912
Saturated Fats	1.177	(0.99, 1.39)	0.057
Added Sugars	0.894	(0.74, 1.06)	0.223
BHEI			
Total fruits	0.787	(0.61, 0.98)	0.044
Whole fruits	0.952	(0.69, 1.31)	0.768
Total vegetables	0.971	(0.33, 2.78)	0.956
Dark green and orange vegetables and legumes	0.770	(0.51, 1.15)	0.211
Total grains	0.715	(0.29, 1.71)	0.454
Whole grains	0.969	(0.73, 1.28)	0.830
Milk and dairy products	0.834	(0.72, 0.96)	0.012
Meats, eggs, and legumes	0.946	(0.66, 1.34)	0.758
Oils		-	
Saturated fat	1.131	(0.97, 1.31)	0.113
Sodium	0.981	(0.82, 1.17)	0.838
Total energies from solid fat, alcohol, and added sugar	0.970	(0.91, 1.03)	0.343
DASH			
Total fruits	0.799	(0.63, 0.99)	0.048
Total Vegetables	1.054	(0.81, 1.36)	0.688
Nuts and Legumes	1.075	(0.86, 1.33)	0.504
Whole grains	0.988	(0.76, 1.28)	0.930
Low-fat dairy	0.755	(0.59, 0.97)	0.021
Sodium	0.921	(0.63, 1.33)	0.667
Red and processed meats	0.820	(0.58, 1.14)	0.243
Sugar-sweetened beverages	0.890	(0.71, 1.11)	0.313

Abbreviations: AHEI, Alternative Healthy Eating Index; HEI-2020, Healthy Eating Index 2020; BHEI, Brazilian Healthy Eating Index Revised; DASH, Dietary Approaches to Stop Hypertension. ^1^ All the models were adjusted for age (years), age squared (years^2^), sex, BMI (kg/m^2^), misreporting (plausible or underreported), per capita family income, self-declared skin color-race (White, Asian or Black, Mixed race, Native), leisure-time physical activity (meet or do not meet WHO recommendation), and use of antihypertensive drugs. The AHEI and DASH models were also adjusted for total energy intake. * A *p*-value < 0.05 was considered statistically significant.

## Data Availability

The data that support the findings of this study are available on request from the corresponding author.

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
