# Peer review of "Associations between Four Diet Quality Indexes and High Blood Pressure among Adults: Results from the 2015 Health Survey of Sao Paulo"

_nutrients, 2024, doi:10.3390/nu16050629_

Round 1
Reviewer 1 Report
Comments and Suggestions for Authors
Well performed study with an excellent draft.
Hypertension could be differently defined for different age groups. Please comment.
Brazilian coke (a variant of Coca Cola) is highly consumed by young people in Brazil (in draft probably assigned under sweetened bevereges). Have you any comment on this habit and its content of sugar/coffein/sopdium?
Diet effects on endothelial function and vascular wall (media/adventitia), atherosclerosis (vascular stiffness) is very complex. By measuring blood pressure in sitting position and then again after standing (within a minute), you can evaluate if the vascular wall reacts to counteract ortostatic hypotension (adequate if the systolic blood pressure not dropping to much and the diastolic increases). In elderly on antihypertensive drugs this could also detect overtreatment with antihypertensive drugs.
The tables and results presentation are fine. The discussion is on a very high level, and I think it could be condensed.
Fruit consumtion is a delicate subject- too much is not good due to high fructose intake - comment?
Green vegetables and diary products, especially cottage cheese/edamer cheese/ vitamin supplemented chicken and red meat contain vitamin K2 - menaquinones which are important for especially media sclerosis/vascular stiffness and indirect the risk for hypertension and vascular disease (MI/stroke) - comment? Do the brazilian people favour fruits for vegetables - comment?
Alcohol intake - wine in high socioeconomic groups contra sweetened alcoholic drinks amongts younger?
Olive oil contra butter/coconut oil intake - comment?
At which age does BMI/subcutaneous/intraabdominal fat (waistline measurement/CT) increase in Brazil - comment?
These are just small queries. Overall this is a fine study and draft!
Reviewer 2 Report
Comments and Suggestions for Authors
Article: Associations between four diet quality indexes and high blood pressure among adults: results from the 2015 Health Survey of Sao Paulo
ID: Nutrients-2863310
The focus of the study is the association between different diet quality indexes and high blood pressure.
It is known that diet influence blood pressure.
The introduction provides sufficient background and it includes relevant references.
The authors could implement introduction with the recent meta-analysis about one of the diet quality indexes studied in their paper (Theodoridis X, Chourdakis M, Chrysoula L, Chroni V, Tirodimos I, Dipla K, Gkaliagkousi E, Triantafyllou A. Adherence to the DASH Diet and Risk of Hypertension: A Systematic Review and Meta-Analysis. Nutrients. 2023 Jul 24;15(14):3261. doi: 10.3390/nu15143261. PMID: 37513679; PMCID: PMC10383418).
The study was performed using a sub-sample of the Health Survey of São Paulo survey. Individual information and societal exposures were self-reported but Dietary Data were rigorously assessed in accordance of National Cancer Institute method and blood pressure was measured and evaluated according to the recommendations.
The research design is appropriate. The results are clearly presented and the conclusions are consistent with the presented arguments. Please check the presentation of the references: some articles showed the year in bold, others not. The paper is well written. I find the present research interesting.
